# The Combination of Anti-CD47 Antibody with CTLA4 Blockade Enhances Anti-Tumor Immunity in Non-Small Cell Lung Cancer via Normalization of Tumor Vasculature and Reprogramming of the Immune Microenvironment

**DOI:** 10.3390/cancers16040832

**Published:** 2024-02-19

**Authors:** Zhan Zhuang, Jinglin Zhou, Minglian Qiu, Jiamian Li, Zhuangheng Lin, Huihan Yi, Xuerong Liu, Changyu Huang, Binghua Tang, Bo Liu, Xu Li

**Affiliations:** 1Key Laboratory of College of First Clinical Medicine, College of First Clinical Medicine, Fujian Medical University, Taijiang Campus, Fuzhou 350001, China; zhuangzhan@fjmu.edu.cn (Z.Z.); qiuminglian@fjmu.edu.cn (M.Q.); ljm996@fjmu.edu.cn (J.L.); lin_zhuangheng@sina.com (Z.L.); yhh_0624@fjmu.edu.cn (H.Y.); liuxueronglxr@163.com (X.L.); cyhuang0520@126.com (C.H.); tangbinghuafjfz@fjmu.edu.cn (B.T.); 2Fujian Key Laboratory of Innate Immune Biology, Biomedical Research Center of South China, College of Life Science, Fujian Normal University Qishan Campus, Fuzhou 350117, China; qsx20211380@student.fjnu.edu.cn

**Keywords:** CD47, CTLA4, tumor immune microenvironment, tumor vascular normalization, non-small cell lung cancer, checkpoint blockade

## Abstract

**Simple Summary:**

Lung cancer stands as the leading cause of cancer-related fatalities both in China and worldwide, underscoring the critical need for effective therapeutic approaches. At present, the primary treatment modalities for non-small cell lung cancer (NSCLC) include chemotherapy, targeted therapy, and immunotherapy. Within this investigation, we demonstrate that the synergistic effects of anti-CD47 and CTLA4 blockade in immunotherapy yield an efficient anti-tumor impact on NSCLC. Furthermore, our study elucidates the underlying mechanisms of this combined therapeutic approach. In conclusion, our research provides compelling evidence supporting the efficacy of the anti-CD47 antibody and anti-CTLA4 antibody combination in treating NSCLC, thereby proposing meaningful avenues for future clinical trials addressing NSCLC.

**Abstract:**

In solid tumors, the formidable anti-tumor impact resulting from blocking the “don’t eat me” signal, arising from CD47–SIRPα interaction, is constrained, especially compared to its efficacy in hematopoietic malignancies. Activating macrophage anti-tumor activity not only necessitates the inhibition of the “don’t eat me” signal, but also the activation of the “eat me” (pre-phagocyte) signal. Intriguingly, the cytotoxic T-lymphocyte-associated antigen 4 (CTLA4) antibody (Ab) has been identified to stimulate Fc receptor-mediated active phagocytes in the tumor microenvironment, thereby generating “eat me” signals. This study postulates that concurrently targeting CD47 and CTLA4 could intensify the anti-tumor effects by simultaneously blocking the “don’t eat me” signal while triggering the “eat me” signal. The experimental data from this investigation confirm that the combined targeting of CD47 and CTLA4 enhances immunity against solid tumors in LLC cell-transplanted tumor-bearing mice. This effect is achieved by reducing myeloid-derived suppressor cell infiltration while increasing the presence of effector memory CD8^+^ T cells, NK1.1^+^ CD8^+^ T cells, and activated natural killer T cells. Meanwhile, combination therapy also alleviated anemia. Mechanistically, the anti-CD47 Ab is shown to upregulate CTLA4 levels in NSCLC cells by regulating Foxp1. Furthermore, targeting CD47 is demonstrated to promote tumor vascular normalization through the heightened infiltration of CD4^+^ T cells. These findings suggest that the dual targeting of CD47 and CTLA4 exerts anti-tumor effects by orchestrating the “eat me” and “don’t eat me” signals, reshaping the immune microenvironment, and fostering tumor vascular normalization. This combined therapeutic approach emerges as a potent strategy for effectively treating solid tumors.

## 1. Introduction

The cell surface glycoprotein cluster of differentiation (CD) 47 is characterized by five transmembrane regions and an immunoglobulin (Ig)-like structural domain, which interacts with the NH2-terminal domain of signal regulatory protein alpha (SIRPα). This interaction between CD47 and SIRP-alpha regulates phagocytosis, and is commonly referred to as the “don’t eat me” signal. *CD47* expression plays a crucial role in preventing autoimmunity and maintaining immune homeostasis in all normal cells, particularly in erythrocytes and hematopoietic stem cells [1,2]. However, the CD47 levels become elevated in cancer cells. Previous research has indicated that CD47 is upregulated in various tumor cells and is positively correlated with an adverse prognosis in breast, gastric, lung, and ovarian cancers [3,4,5,6]. The upregulated CD47 in tumor cells interacts with SIRPα, which is prominently expressed on the surface of myeloid cells, triggering “don’t eat me” signals. This interaction enables tumor cells to evade innate immunity [2]. 

Clinical trials have confirmed that the potency and efficacy of CD47/SIRPα-based therapy, utilizing CD47-blocking monoclonal antibodies (Abs) or SIRPα-fragment crystallizable (Fc) fusion proteins, is a potent and efficacious approach for treating both solid tumors and hematologic malignancies. However, the current low responsiveness of oncology patients to CD47 monotherapy and its severe post-treatment adverse effects, such as hemolysis and thrombocytopenia, limit the therapeutic efficacy of CD47 [2]. Given the intricate nature of the tumor immune microenvironment (TIME), blocking a single signaling pathway on a subset of immune cells can only yield a restricted or minor effect. Therefore, exploring other drugs in combination with anti-CD47 Ab therapy is essential to avoid systemic adverse effects and to improve efficacy.

As a crucial immune checkpoint, the cytotoxic T-lymphocyte-associated antigen 4 (CTLA-4) blockade Ab has demonstrated encouraging anti-tumor effects in various cancers, including lung cancer, melanoma, and colon cancer [7,8,9]. In an independent study, anti-CTLA4 Ab was found to enhance the anti-tumor effect by altering the ratio of effector T cell (T_eff_) to regulatory T cell (T_reg_) in the tumor microenvironment (TME) in human Fc-gamma receptors (FcγRs)-treated mouse models inoculated with MCA205 cells [10]. However, studies also revealed that anti-CTLA4 Ab could enhance the anti-tumor activity depending on the activation of Ab-dependent cellular phagocytosis (ADCP). This occurs when the Ab binds to CTLA4 on the surface of the targeted cells and triggers human Fc receptors (FcRs) expressed on macrophages to generate “eat me” signals [10,11,12].

The interaction between CD47 and SIRPα, producing the “don’t eat me” signal, may limit the effectiveness of the “eat me” signal generated through the activation of FcRs by anti-CTLA4 Ab. A hypothesis was proposed to develop a combination therapeutic approach to enhance the anti-tumor potential. This involves using anti-CD47 Ab to block the CD47–SIRPα interaction and the subsequent generation of the “don’t eat me” signal in the TME, along with anti-CTLA4 Ab to activate the “eat me” signal. To evaluate this hypothesis, a mouse model of non-small cell lung cancer (NSCLC) with subcutaneous transplant tumors was established, and the anti-tumor effects of the combined therapy using anti-CD47 Ab and anti-CTLA4 Ab were assessed. This combination approach may provide new treatment options for NSCLC and shed light on potential targeted treatment mechanisms.

## 2. Materials and Methods

### 2.1. Cell Lines and Culture Conditions

The murine Lewis lung carcinoma (LLC) cell line, obtained from the Cell Bank of the Chinese Academy of Sciences (Shanghai, China), underwent identification through short tandem repeat analysis. Additionally, the NCI-H1975 and NCI-H1650 cell lines (derived from human samples) were graciously donated by the Biomedical Research Center of South China at Fujian Normal University (Fuzhou, China). These cell lines were cultured in a medium (HyClone, Seattle, WA, USA) supplemented with 10% fetal bovine serum (FBS, PAN-Biotech, Aidenbach, Germany) in an incubator with 5% CO_2_, with passaging occurring within a period of less than 6 months.

The LLC-Foxp1^−/−^, NCI-H1975-Foxp1^−/−^, and NCI-H1650-Foxp1^−/−^ cells were generated using clustered regularly interspaced short palindromic repeats (CRISPR)-associated protein 9 (CRISPR-Cas9) genome engineering. The specific guide RNA sequences used were as follows: Human *Foxp1* SgRNA1: AACCACTTACTAGAGTGCGG; Human *Foxp1* SgRNA2: GACGCCGGCCGTGGACATCG; Mouse *Foxp1* SgRNA1: TGTCATCATAGCCACTGACA; Mouse *Foxp1* SgRNA2: CAACCACTTACTAGAGTGCG). All the tested cell lines demonstrated no evidence of mycoplasma contamination.

### 2.2. Tumor Inoculation, Treatment, and Measurement

LLC cells were cultured in Dulbecco’s modified Eagle’s medium (DMEM) supplemented with 10% FBS. Tumors were induced in female C57BL/6 mice by subcutaneously injecting LLC cells (1 × 10^6^ per mouse) in 100 μL phosphate-buffered saline (PBS) into the right shoulder. Specific pathogen-free (SPF) female C57BL/6 mice, aged 6–8 weeks and weighing 18–25 g, were procured from Shanghai Slac Laboratory Animal (Experimental Animal License: SCXK (Shanghai), #2022-0004; Shanghai, China). Prior to the experiment, mice were fully anesthetized with 2–3% isoflurane for 1–2 min, leading to rapid loss of consciousness. Following the animals’ demise, 2–3% isoflurane was continued for 1 min, and euthanasia was then performed using the cervical dislocation method.

Seven days after tumor cell inoculation, mice were subjected to treatment with anti-mouse CD47 Ab (MIAP410; BioXcell, West Lebanon, NH, USA), anti-mouse CTLA4 Ab (BE0131; BioXcell, West Lebanon, NH, USA), control IgG anti-mouse (CP160; BioXcell, West Lebanon, NH, USA), and anti-mouse CD4 Ab (BE0003-1; BioXcell, West Lebanon, NH, USA) alone or in combination (i.e., anti-CD47 Ab with anti-CTLA4 Ab or anti-CD47 Ab with anti-CD4 Ab), respectively. The allocation of mice to treatment groups occurred randomly once the tumor diameter reached ~5 mm (*n* = 5–8). Throughout the treatment period, Abs diluted with PBS (10 mg/kg) (anti-CD47 Ab, anti-CTLA4 Ab, and anti-CD4 Ab) were administered intraperitoneally three times a week for 24 days. Tumor measurements were taken with vernier calipers, and tumor size was calculated using the formula: π/6 × length × width × height. All experimental procedures in this research were conducted in accordance with the Guide for the Care and Use of Laboratory Animals. Approval was obtained from the Ethics Committee for Animal Experimentation of Fujian Medical University.

### 2.3. Fluorescence-Activated Cell Sorting (FACS)

To assess tumor-infiltrating lymphocytes (TILs), mice were initially inoculated with LLC cells to establish tumors and subsequently treated with either anti-CD47 Ab or anti-CTLA4 Ab alone or in combination, as previously described. On day 24, tumors were excised, minced, and subjected to treatment with Hank’s balanced salt solution (HBSS). The samples were then cultured with 15 μg/mL collagenase and 1.5 mg/mL hyaluronidase, agitated for 30 min at 37 °C in a shaker. Following digestion, these tissues were filtered through a 70 μm cell filter and washed twice with PBS/0.5% bovine serum albumin (BSA). Erythrocytes were lysed using an erythrocyte lysis buffer (#555899; BD Biosciences, Franklin Lakes, NJ, USA), and the cell precipitate was resuspended in 11 mL 40% Percoll (GE Healthcare, Chicago, IL, USA) in Roswell Park Memorial Institute (RPMI), covered with 3.5 mL of a 70% Percoll gradient in a 15 mL conical tube. After centrifugation at 800× *g* for 30 min, lymphocytes collected from the gradient interface were rinsed with ice-cold RPMI 1640 (10 mL) and resuspended. Lymphocytes were stained with a mixture of Abs, including the following obtained from BD Biosciences: Live/Dead-FVS780-APC-Cy7 (#565388), anti-mouse CD45-BUV395 (#564279), anti-mouse CD3-AF700 (#561388), anti-mouse MHCII-BV605 (#563413), anti-mouse CD62L-BV650 (#564108), anti-mouse CD127-PE-Cy7 (#560733), anti-mouse F4/80-AF647 (#565853), anti-mouse CD86-BUV737 (#741737), anti-mouse CD4-BUV496 (#612952), anti-mouse PD-L1-BV421 (#564716), anti-mouse CD80-BV711 (#740698), anti-mouse PD-1-BB515 (#566832), anti-mouse CD8-BUV805 (#612898), anti-mouse NK1.1-PE-CF594 (#562864), anti-mouse Ly-6G-BV480 (#746448), anti-mouse CD25-BB700 (#566498), anti-mouse CD11b-BUV563 (#741242), and anti-mouse CD11c-PE (#557401); and the following from BioLegend (San Diego, CA, USA): anti-mouse Ly-6C-BV785 (#128041) and anti-mouse CD69-PE-Cy5 (#104510). Stained cells were observed using BD FACSymphony (BD Biosciences), and the FACS data were analyzed using the software (FlowJo 10.8.1) included with the instrument.

### 2.4. Immunohistochemistry (IHC) and Immunofluorescence (IF)

The excised tumor tissues were fixed using 4% paraformaldehyde and embedded in paraffin. Sections sliced to 4 μm thickness were dewaxed, hydrated, and subjected to high-temperature antigen retrieval. After blocking with BSA to minimize non-specific protein interactions, the samples underwent overnight incubation with primary Abs, followed by 30 min incubation with secondary Abs. Nuclei were stained using 4’,6-diamidino-2-phenylindole (DAPI) and counter-stained with hematoxylin. The sections were then sealed.

For identification, vascular endothelial cells were labelled using anti-CD31 Ab (GB12064; Servicebio, Wuhan, China), vascular smooth muscle cells (VSMCs) were identified using anti-α smooth muscle actin (α-SMA) Ab (ab124964; Abcam, Cambridge, UK), and capillary pericytes were labeled using anti-neural/glial antigen-2 (NG2) Ab (Cy3-conjugated polyclonal Ab, AB5320C3; Millipore, Billerica, MA, USA). The protein levels of CTLA4, Foxp1, and Ki-67 in tumor tissues were assessed using anti-CTLA4 Ab (BM5388; Boster, Pleasanton, CA, USA), anti-Foxp1 Ab (ab134055; Abcam), and anti-Ki-67 Ab (ab16667; Abcam). Subsequently, samples were incubated using fluorescence-coupled anti-mouse or anti-rabbit secondary Abs (Alexa 488 and Alexa 594; Thermo Fisher Scientific, Waltham, MA, USA). Sections were sealed using ProLong™ Gold antifade mountant containing DAPI (Thermo Fisher Scientific) to counter-stain the nuclei. The total number of vessels covered by pericytes and VSMCs was quantified in five randomly selected areas using a microscope (×200, 50 μm). Data analysis was performed using the ImageJ v1.57 software (National Institutes of Health, Bethesda, MA, USA) and Photoshop v25.1 software (Adobe, San Jose, CA, USA).

### 2.5. Western Blot Analysis

Following various treatments, the cells were washed with ice-cold PBS and then collected through gentle scraping. Total proteins were extracted by lysing the cells with radioimmunoprecipitation assay lysis buffer. The samples were subjected to 12.5% tricine-sodium dodecyl sulfate-polyacrylamide gel electrophoresis for protein isolation. The separated proteins were subsequently transferred onto polyvinylidene fluoride membranes. These membranes were incubated with anti-CTLA4 Ab (diluted 1:1000; Thermo Fisher Scientific), anti-Foxp1 Ab (diluted 1:1000; Abcam), and anti-CD47 Ab (diluted 1:1000; Abcam). Afterwards, the membranes were further incubated with IRDye 800CW or 680 LT secondary Ab (1:1000). Blots were visualized using the OdysseyCLx Western blot detection system (Westburg, Leusden, The Netherlands). Glyceraldehyde-3-phosphate dehydrogenase (GAPDH) (diluted 1:10,000; Abcam) served as an endogenous control. Data analysis was conducted using the ImageJ v1.57 software (US National Institutes of Health) and Photoshop v25.1 software (Adobe).

### 2.6. Routine Blood Test in Mice

On experiment day 24, mice were subjected to isoflurane inhalation anesthesia. Subsequently, 70–100 μL blood was collected from each mouse using microhematocrit tubes via retro-orbital bleeding. The red blood cell (RBC) count, white blood cell (WBC) count, and hemoglobin (Hb) concentration were determined using an automatic hemocytometer (BC-5000vet; Mindray, Shenzhen, China).

### 2.7. Statistical Analysis

The sample sizes were not pre-determined using statistical methods. Mice were randomly assigned to each treatment group and, when possible, mixed in cages. Throughout the experiments, researchers were blinded to the treatment protocols when assessing tumor size with calipers. Data were analyzed using the Prism v5.0 software (GraphPad, San Diego, CA, USA) and presented as mean ± standard error. The Kaplan-Meier method was employed to estimate the median overall survival (OS). The *p*-values were calculated to assess differences using Student’s *t*-test or one-way analysis of variance, with the significance levels determined as follows: “ns” describes no significance, * *p* < 0.05, ** *p* < 0.01, *** *p* < 0.001. Survival was statistically analyzed using Cox proportional risk models, with the hazard ratios (HRs) and 95% confidence intervals (CIs) calculated.

## 3. Results

### 3.1. Dual CD47 and CTLA4 Blockade Impedes Tumor Growth and Improves Survival in NSCLC-Bearing Mice

First, a subcutaneous transplantation tumor model of NSCLC cells in mice was established, and this model was treated with a combination of Abs targeting CD47 and CTLA4. The efficacy of the combination therapy was then evaluated. Mice with NSCLC tumors ranging between 60 and 80 mm^3^ in size were administered anti-CD47 Ab (at a dosage of 10 mg/kg thrice/week), anti-CTLA4 Ab (at a dosage of 10 mg/kg thrice/week, or a combination of both.

Significant growth delay was observed in the tumor growth for both the anti-CD47 Ab group and the anti-CTLA4 Ab group compared to the control group. However, the tumor growth delay was more pronounced when both anti-CD47 Ab and anti-CTLA4 Ab were administered together compared to the single-agent treatment groups (Figure 1A). Furthermore, there was a notable increase in the median OS in the combination therapy group compared to the monotherapy or control groups (Figure 1B). These findings suggest that the administration of both anti-CD47 Ab and anti-CTLA4 Ab can restrict tumor growth and improve OS in mice with tumors.

### 3.2. Alteration of Immune Cell Reprogramming in TME and Potentiation of Immunotherapy Response in NSCLC-Bearing Mice by a Combination of Anti-CD47 Ab and Anti-CTLA4 Ab

The TME, comprising mainly innate and adaptive immune cells along with other non-immune cell types, plays a crucial role in tumorigenesis and anti-tumor immunity [13,14,15]. Therefore, we initially investigated the impact of combination therapy on lymphocyte infiltration in tumors. After 24 days of combination therapy (anti-CD47 Ab and anti-CTLA4 Ab), the mice were euthanized, and the tumors were dissected for digestion. Single-cell suspensions were prepared using flow cytometry for a comprehensive analysis of changes in the distribution of each subpopulation of tumor-infiltrating immune cells. The proportion of tumor-infiltrating CD45^+^ immune cells was elevated in response to the combination group (Appendix A) compared to the single-agent treatment groups (anti-CD47 Ab and anti-CTLA4 Ab alone). However, in both the monotherapy group and the combination therapy group, the CD3^+^ T cell infiltration of the tumors increased, but the infiltration number was not significantly different between the monotherapy group and the combination therapy group. The trend of CD8^+^ T cell infiltration was consistent with that of CD3^+^ T cell infiltration (Appendix A). Additionally, it was found that the anti-CD47 Ab group and combination therapy promoted CD4^+^ T cell infiltration in the tumors compared with the anti-CTLA4 Ab group and the control group (Appendix A).

Next, the impact of combination therapy on the infiltration of CD8^+^ T cell subsets in the tumors was evaluated. Flow cytometric analysis highlighted that both the monotherapy and combination therapy promoted the infiltration of activated CD8^+^ T cells (CD69^+^, CD8^+^ T cells) in the tumors compared to the control group. Moreover, it was found that the effector memory CD8^+^ T cells (CD62L^−^ CD127^+^ and CD8^+^ T cells) in the combination therapy group showed significantly increased infiltration (Figure 2A), with non-significant infiltration in the other subsets. Meanwhile, the infiltration of exhausted CD8^+^ T cells was reduced in the combination therapy and anti-CTLA4 groups compared with the control group; however, such alterations were not observed in the anti-CD47 Ab groups (Figure 2B). NK1.1^+^ CD8^+^ T cells, a subtype of CD8^+^ T cells, are derived from CD8^+^ T cells during the initiation process and act as effector cells. They provide innate immune protection and stimulate inflammatory response primarily by secreting IFN-γ and granzyme B [16,17]. NK1.1^+^ CD8^+^ T cells, a subtype of CD8^+^ T cells, increased significantly after monotherapy (anti-CTLA4 Ab) and combination therapy, with no such alterations observed in the monotherapy (anti-CD47 Ab) and control groups (Figure 2C).

Flow cytometry was employed to assess changes in the bone marrow cell population after treatment. No notable change was observed in tumor macrophage infiltration with either combination therapy or monotherapy (Appendix A). However, the combination therapy increased M1-phenotype macrophage (F4/80^+^ CD86^+^ MHCII^+^) infiltration in the tumors, while such alterations were not observed in the other groups (Figure 2D). Additionally, it was observed that the proportion of myeloid-derived suppressor cells (MDSCs) infiltration decreased considerably in the combination therapy and monotherapy groups compared to the control group. Furthermore, in the treatment groups (combination and monotherapy), MDSCs were reduced in the combination group compared to the anti-CD47 group (Appendix A).

Finally, the alterations in natural killer (NK) cell and natural killer T (NKT) cell infiltration were examined using flow cytometry after the combination therapy. NK cells were considerably elevated after combination therapy compared to the other groups; however, the proportion of NKT cell infiltration did not change significantly (Appendix A). Importantly, after combination therapy, the number of activated NKT cells accounted for a higher percentage of the total number of NKT cells compared to the monotherapy groups and control group. (Figure 2E). The above experimental results suggest that TIME reprogramming plays a critical role in mediating the anti-tumor effect of the combination of anti-CD47 Ab and anti-CTLA4 Ab.

### 3.3. Upregulation of CTLA4 Level in NSCLC Cells by Anti-CD47 Ab via Regulation of Foxp1

There is often mutual regulation among immune checkpoints in the TIME [18,19,20], which prompted us to further evaluate the PD-1, PD-L1, and CTLA4 protein levels in the tumor tissues using IHC after anti-CD47 Ab treatment. The IHC results (Figure 3A) revealed that CTLA4 was upregulated in the tumor tissues compared to the controls, but the levels of PD-1 and PD-L1 were not significantly different (Appendix A). Next, staining with Abs against KI-67 and CTLA4 was employed to assess the ratio of NSCLC cells displaying the CTLA4 level. The IF double staining assay (Figure 3B) demonstrated that the CTLA4 level in the tumor tissues was mainly concentrated in tumor cells after anti-CD47 Ab treatment. Additionally, the tumor tissue was ground and filtered into a single-cell suspension, and the CTLA4 protein level was detected using Western blot. As shown in Appendix A, the level of CTLA4 was significantly increased after treatment with anti-CD47 Ab compared to the wild-type group. This result further confirmed that anti-CD47 Ab increased the CTLA4 levels in tumor cells.

Subsequently, the molecular mechanism of the promotive effect of anti-CD47 Ab on the cellular level of CTLA4 was explored. Foxp1 belongs to the Foxp transcription factor subfamily (Foxp1–4), which regulates differentiation in various cell types [21]. Based on CRISPR-CAS9 screening, Foxp1 was found to be one of the important regulatory molecules of the CTLA4 level in T cells, indicating that the CTLA4 level was significantly downregulated in T_reg_ cells with Foxp1 deficiency [21,22]. As a result, further examination was conducted to investigate whether Foxp1 contributed to the upregulation of the CTLA4 level in NSCLC treated with anti-CD47 Ab. First, using IHC, we identified that the the Foxp1 level was upregulated in the anti-CD47-treated tumor tissues compared to the control group. In addition, we confirmed this finding via Western blot (Figure 3C,D). Subsequently, we investigated whether Foxp1 regulates the level of CTLA4 in NSCLC cell lines. The results showed that the CTLA4 level was downregulated in *Foxp1*^−/−^ NSCLC cells (*Foxp1* gene knocked out cell lines), as highlighted by Western blot (Figure 3E).

To explore the effect of Foxp1 on the anti-CD47 Ab regulation of the CTLA4 level, we constructed a subcutaneous transplantation tumor model with Foxp1-deleted LLC cells and wild-type LLC cells, respectively, in mice, and treated them with anti-CD47 Ab. We assessed changes in the Foxp1 expression levels using IHC in the subcutaneous transplantation tumor model with Foxp1-deleted LLC cells and wild-type LLC cells, respectively. As shown in Appendix A, the Foxp1 expression levels were significantly reduced in the Foxp1-deleted tumor model compared to the wild-type tumor model. We evaluated the protein level of CTLA4 using IHC and Western blot. The data showed that the CTLA4 level was reduced in the Foxp1-deleted tumor model compared to the wild-type tumor model (Figure 3F). Meanwhile, the CTLA4 level in the single-cell suspension obtained from the tumor tissue was ground and filtered, and it was detected with immunoblots (Appendix A). As described above, these experimental results imply that Foxp1 is one of the important regulators of anti-CD47 Ab in promoting the upregulation of the CTLA4 level in NSCLC.

### 3.4. Anti-CD47 Ab Promotes Vascular Normalization of Tumors

It has previously been reported that the CD47 level is elevated in mouse models of atherosclerosis. CD47 Ab blockade could reverse malignant cell resistance to programmed cell removal (PrCR), allowing the clearance of diseased vascular tissue and promoting its normalization to ameliorate atherosclerosis [23]. Nevertheless, the function of CD47 in tumor angiogenesis is not yet well understood. To investigate the functional role of targeting CD47 in tumor angiogenesis, anti-CD47 Ab was injected into C57BL/6 mice bearing tumors of LLC cells using subcutaneous transplantation tumor models. The microvessel density (MVD) and morphology of tumors on CD31-stained tumor sections were assessed through confocal fluorescence microscopy, followed by quantitative image analysis. The MVD in the anti-CD47 Ab-treated mice was elevated, with increased branching, compared to the controls (Figure 4A). These data indicate that anti-CD47 Ab promotes angiogenesis within the tumor.

Pericytes and VSMCs supporting the vascular endothelium are significant regulators of vascular maturation and function [24,25]. To explore how anti-CD47 Ab impacts tumor vascular wall cell coverage, IF double staining of α-SMA and CD31 was used, and it was found that the percentage of α-SMA-positive wall cells covering the blood vessels was increased in the anti-CD47 Ab-treated group compared to the control group (Figure 4B). Consistent with this observation, the percentage of chondroitin sulfate proteoglycan (NG2)-positive pericytes that cover vessels was also higher in the anti-CD47 Ab-treated tumors than in the control group (Figure 4C). The above data indicate that anti-CD47 Ab augments durable angiogenesis and normalization.

In addition, we evaluated the effect of combining Anti-CD47 and Anti-CTLA4 therapy on tumor vascular normalization. Compared with the single-agent therapy (Anti-CD47), the combination therapy slightly increased the coverage of α-SMA wall cells (Appendix A). Meanwhile, compared with the monotherapy (Anti-CTLA4), the combination therapy improved the vascular coverage of NG2 perivascular cells (Appendix A). These results indicate that tumor vascular normalization is one of the important mechanisms of combination therapy.

### 3.5. Promotion of Tumor Vascular Normalization by Anti-CD47 Ab via Enhancement of CD4^+^ T Cell Tumor Infiltration

A recent report revealed that vascular normalization and immunostimulatory reprogramming might regulate each other, highlighting the key role of infiltrating CD4 T cells in promoting tumor vascular normalization [26]. IHC (Appendix A) and flow cytometric analysis (Figure 5A) demonstrated that anti-CD47 Ab treatment promoted CD4^+^ T cell infiltration in tumors, in contrast to the non-significant alteration in the control group. Additionally, anti-CD47 Ab was found to enhance the infiltration of activated CD4^+^ T cells within the tumors (Appendix A).

Next, tumors of LLC cells were established in C57BL/6 mice, and these mice were given control IgG, anti-CD47 Ab alone, or a combination of anti-CD47 Ab with anti-CD4 Ab. It was investigated whether anti-CD47 Ab regulated intra-tumor vasculature and functioned through increased CD4^+^ cell infiltration. The impact of the CD4^+^ T cells on tumor vascular normalization in the anti-CD47 Ab treatment was analyzed using IF double staining of CD31, α-SMA, and NG2 (Figure 5B–D). The results showed no significant changes in the CD31 level in the combination therapy group compared to the anti-CD47 Ab group. However, both the α-SMA-positive wall cells and NG2-positive pericytes exhibited decreased vascular coverage. Moreover, the tumor volume was slightly increased in the combination therapy group compared to the anti-CD47 Ab group (Figure 5E).

Thus, the above experimental results indicate that in the context of the anti-CD47 Ab treatment of tumors, anti-CD47 Ab promoted tumor vascular normalization by increasing the infiltration of CD4^+^ T cells. Furthermore, tumor vascular normalization may also be an important mechanism that increases the efficacy of anti-CD47 Ab treatment of tumors.

### 3.6. Alleviation of Anemia in NSCLC-Bearing Mice by Combined Targeting of CD47 and CTLA4

In recent years, most CD47 monoclonal Ab-based clinical trials have been terminated [27]. Severe hemolytic reactions caused by CD47 monoclonal Ab-induced blood coagulation are a major issue leading to clinical failure [28]. We evaluated the impact of combined therapy targeting CD47 and CTLA4 on the blood cells of NSCLC-bearing mice. Blood from the mice was extracted through eyeball extirpating, and a comprehensive analysis of mouse blood cells was performed using a fully automatic hemocytometer. The combination therapy elevated the Hb concentration and mean corpuscular hemoglobin concentration (MCHC) in the blood (Figure 6A,B) in comparison to monotherapy (anti-CD47 Ab). The WBC, RBC, and platelet counts in the blood of the mice were also measured, but neither combination therapy nor monotherapy significantly altered these cell ratios (Figure 6C–E). Interestingly, it was found that combination therapy enhanced the proportion of eosinophils in the blood (Figure 6F). Therefore, based on the above data, it can be concluded that combination therapy can alleviate anemia by increasing the Hb concentration and ameliorating the hemolytic reaction induced by CD47 monoclonal Ab.

## 4. Discussion

The interaction between CD47 and SIRPα triggers “don’t eat me” signals from macrophages, inhibiting phagocytosis and enabling tumor cells to evade immune surveillance [1]. However, CD47 blockade alone is insufficient to fully stimulate macrophage-mediated anti-tumor activity as it requires concurrent “eat me” signals [29]. There are alternative anti-macrophage pathways beyond the CD47–SIRPα interaction [30]. Given the complexity of the TIME, targeting a single signaling pathway on specific immune cells may have limited effects [31,32]. Therefore, actively exploring effective targets to co-block CD47 is crucial to fully activate macrophages for the phagocytosis of tumor cells.

CTLA4, an important immune checkpoint blocker (ICB), is primarily expressed in immune cells, but has also been found in tumor cells [19,33,34,35]. Anti-CTLA4 Ab has demonstrated promising anti-tumor effects across various cancer types [7,9]. By binding to CTLA4 on target cells, anti-CTLA4 Ab engages FcRs on macrophages, generating “eat me” signals that trigger Ab-dependent cell-mediated phagocytosis—a significant mechanism contributing to the anti-tumor effects of CTLA4 Ab [10,11,12,36]. This study hypothesized that blocking the “don’t eat me” anti-macrophage phagocytic signaling with anti-CD47 Ab and activating the “eat me” macrophage phagocytic signaling with anti-CTLA4 Ab could enhance anti-tumor efficacy. The research utilized a subcutaneous transplantation tumor model of NSCLC in mice, observing significant anti-tumor effects with the combination therapy targeting CD47 and CTLA4. The study also confirmed the alteration of the TIME upon combination therapy and noted a partial alleviation of anemia. Mechanistically, the study found that targeting CD47 promoted the upregulation of CTLA4 levels in NSCLC cells by regulating the transcription factor Foxp1. Furthermore, targeting CD47 was validated to promote the normalization of tumor vasculature by increasing the infiltration of CD4^+^ T cells.

Therapies targeting CD47, including anti-CD47 monoclonal Ab and bispecific Abs for CD47, have shown promising outcomes in clinical studies. However, severe hemolytic reactions and drug resistance pose challenges to the anti-tumor efficacy of CD47-targeting therapies [2]. In NSCLC-bearing mouse models, a significant delay in tumor growth was observed with anti-CD47 Ab and anti-CTLA4 therapy alone compared to the control group. Notably, the combination of anti-CD47 Ab and anti-CTLA4 Ab resulted in a more substantial delay in tumor growth than either of the single-agent treatments.

Further analysis revealed that the combination therapy also improved the median survival in the animal models compared to the monotherapy group (anti-CD47 Ab or anti-CTLA4 Ab alone). These findings demonstrated the significant anti-tumor effect of the combined targeting of CD47 and CTLA4. Moreover, the combination therapy increased the blood Hb concentration and MCHC and elevated the eosinophil proportions compared to the monotherapy. These results indicated that the combination therapy targeting CD47 and CTLA4 alleviated the anemia caused by CD47 monoclonal Ab in NSCLC-bearing mice.

The TME plays a crucial role in mediating the impact of therapy on tumors, and it can be categorized into an activating TIME with anti-tumor capacity and a suppressive TIME with tumor-supporting capacity [37]. While previous studies on TIME in therapies targeting CD47 and combined therapy targeting CD47 have focused mainly on the effect of CD8^+^ T cells and macrophages on therapeutic outcomes [1,32,38,39], this study used flow cytometry to comprehensively examine the impact of combined therapy targeting CD47 and CTLA4 on immune cell infiltration in the TME. The study found that the combined targeting of CD47 and CTLA4 reduced the infiltration of MDSCs compared to single-agent CD47 therapy. Additionally, the combined targeting of CD47 and CTLA4 increased the infiltration of effector memory CD8^+^ T cells, M1 macrophages, and activated NKT cells compared to single-agent therapy (anti-CD47 or anti-CTLA4). These findings illustrate the anti-tumor activity of activating TIME through combination therapy while limiting the suppressive immune cell population that promotes tumor development. Surprisingly, the analysis of CD8^+^ T cell subtypes revealed a significant increase in intra-tumor NK1.1^+^ CD8^+^ T cell infiltration after anti-CTLA4 Ab and combination therapy, unveiling a novel anti-CTLA4 Ab-dependent anti-tumor mechanism.

CTLA4 functions as a regulator that limits the activity of activated T cells, identified as a crucial immune checkpoint in the regulation of T cell activation. While previous studies have predominantly focused on the role of CTLA4 in immune cells, emerging evidence suggests its level in tumor cells as well [19,40]. In this study, it was observed that targeting CD47 led to an upregulation of CTLA4 levels in tumor cells, as evidenced by the IHC and IF assays. Furthermore, the IHC results revealed an upregulation of the Foxp1 level in the tumor tissues after anti-CD47 Ab treatment. Employing CRISPR-CAS9 technology to knock out *Foxp1* in NSCLC cells, the Western blot analysis demonstrated the downregulation of the CTLA4 level in Foxp1-deficient NSCLC cells. Subsequently, in a subcutaneous transplantation tumor model using Foxp1-deficient LLC cells treated with anti-CD47 Ab, it was observed that Foxp1 deletion partially prevented the increase in the CTLA4 level induced by anti-CD47 Ab treatment. These findings establish that anti-CD47 Ab promotes the upregulation of CTLA4 levels in NSCLC cells through the regulation of Foxp1, providing a theoretical foundation for combined therapy targeting CD47 and CTLA4 in NSCLC.

Thrombospondin-1, as one of the ligands of CD47, typically inhibits tumor angiogenesis by blocking endothelial cell proliferation and chemotaxis [41,42]. Surprisingly, in this study, targeting CD47 was found to promote tumor angiogenesis, indicating a deviation from the expected anti-angiogenic effect. Through the reversal of malignant cell resistance to PrCR, CD47 Ab blockade facilitated the clearance of diseased vascular tissue and promoted its normalization, ultimately ameliorating atherosclerosis [23]. Using IF double staining of vascular endothelial cells with CD31, α-SMA, and NG2, it was discovered that targeting CD47 significantly increased the coverage of α-SMA^+^ wall cells and NG2^+^ pericytes in tumor vascular endothelial cells. This suggests a potential reciprocal regulation between immunostimulatory reprogramming and vascular normalization [26]. Recent studies have also indicated that immune checkpoint blockade can mediate the normalization of tumor vasculature via CD4^+^ T cells in an IFN-γ-dependent manner [26]. The current work elaborated that targeting CD47 promoted the infiltration and activation of CD4^+^ T cells in tumors. Moreover, in a subcutaneous transplantation tumor mouse model, the depletion of CD4^+^ T cells in mice using anti-CD4 Ab alongside anti-CD47 Ab therapy significantly diminished the effect of anti-CD47 Ab in promoting tumor vascular normalization. Additionally, it was observed that the depletion of CD4^+^ T cells in mice while targeting CD47 led to a slight elevation in the tumor volume compared to targeting CD47 alone. These results suggest that CD47 promotes tumor vascular normalization by increasing CD4^+^ T cell tumor infiltration, indicating that tumor vascular normalization is a crucial mechanism contributing to the anti-tumor effects of anti-CD47 Ab. Furthermore, the effect of combination therapy on tumor vascular normalization was evaluated. Compared with single-agent therapy (anti-CD47), the combination therapy slightly increased the coverage of NG2 perivascular cells. Simultaneously, compared with monotherapy (anti-CTLA4), combination therapy improved the vascular coverage of α-SMA wall cells. These results indicate that tumor vascular normalization is one of the important mechanisms of combination therapy.

## 5. Conclusions

In summary, our study demonstrates the efficacy of combination therapy involving anti-CD47 Ab and anti-CTLA4 Ab in NSCLC models. This combination approach selectively activates anti-tumor immune cells within the TIME of NSCLC while concurrently suppressing the immunosuppressive cell population. Notably, we emphasize that anti-CD47 Ab promotes the upregulation of CTLA4 levels in NSCLC cells through the regulation of Foxp1. Importantly, an observed significant increase in NK1.1^+^ CD8^+^ T cell infiltration was responsive to anti-CTLA4 Ab monotherapy or combination therapy. Furthermore, our study substantiates that anti-CD47 Ab contributes to tumor vascular normalization by enhancing CD4^+^ T cell infiltration, highlighting the role of tumor vascular normalization in the anti-tumor effect of CD47 targeting. These findings have important implications for future clinical trials involving these relevant agents in the treatment of NSCLC and other cancers.

## Figures and Tables

**Figure 1 cancers-16-00832-f001:**
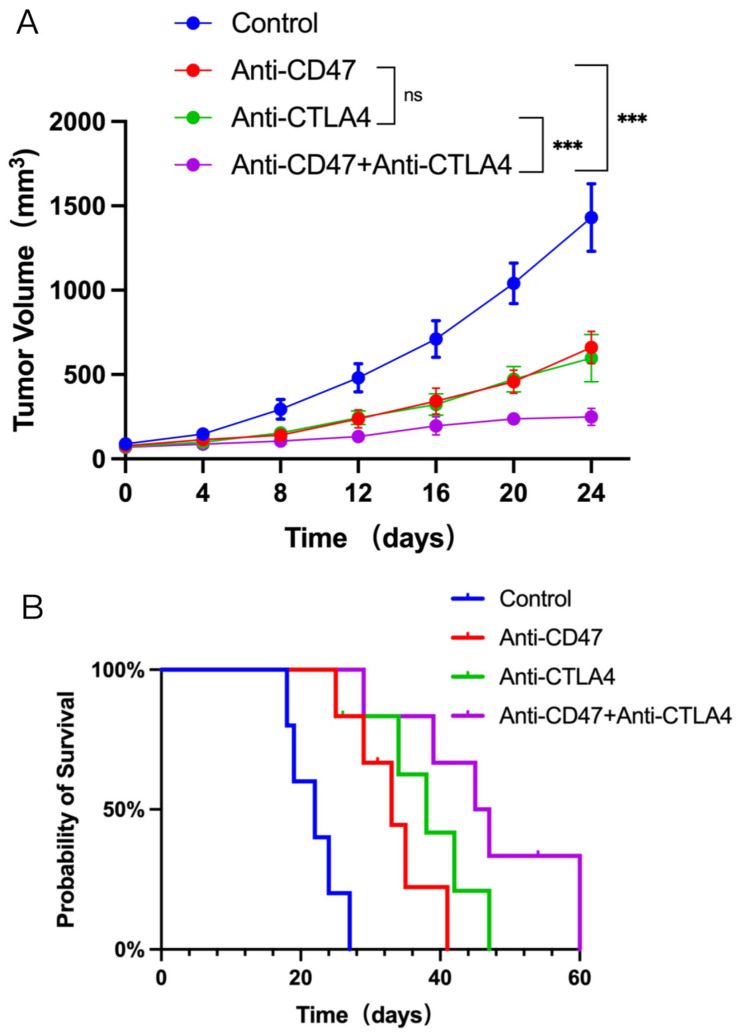
Efficacy of combination therapy of anti-CD47 antibody (Ab) and anti-CTLA4 Ab in non-small cell lung cancer (NSCLC) mouse models. C57BL/6 mice (*n* = 5–8 per group) bearing tumors of LLC cells were treated with control rat immunoglobulin G (IgG) (10 mg/kg) or administered with anti-CD47 Ab (10 mg/kg) and anti-CTLA4 Ab (10 mg/kg, three times weekly) alone or in combination. These treatments were administered intraperitoneally three times a week for 24 days. Tumor growth (**A**) and survival rates (**B**) were closely observed throughout the experimental period. (**A**) In non-small cell lung cancer (NSCLC) mouse models, tumor volume was presented after treatment with Anti-CD47 Ab, Anti-CTLA4 Ab or Anti-CD47 Ab plus Anti- CTLA4 Ab. (**B**) Kaplan-Meier survival distribution graph illustrating the impact of different treatments on mouse survival. Quantitative analysis of tumor volume is displayed as mean ± standard deviation (SD). (*n* = 5 per group). Ns, no significance; *** *p* < 0.001.

**Figure 2 cancers-16-00832-f002:**
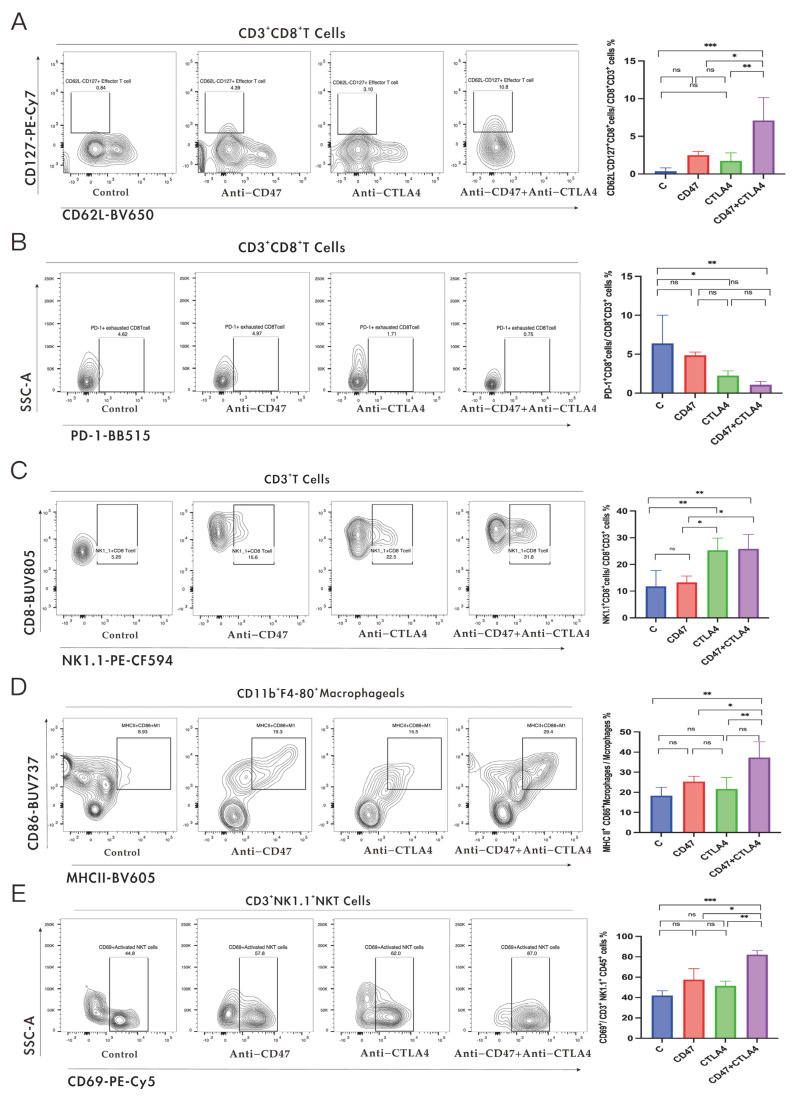
Effect of dual CD47 and CTLA4 blockade on immune stimulation in non-small cell lung cancer NSCLC-bearing mice. Tumors of LLC cells were established in C57BL/6 mice, and these mice were given control IgG, or anti-CD47 Ab and anti-CTLA4 Ab alone or in combination. Analysis of tumor-infiltrating lymphocyte (TIL) subtype (represented as fractions of CD45^+^ cells) in tumors of LLC cells, measured by flow cytometry at day 24. Percentages of memory effector CD8^+^ T cells (CD62L-CD127+ CD8^+^ T cells) (**A**), infiltrating exhausted CD8^+^ T cells (**B**) and NK1.1+ CD8^+^ T cells (**C**), M1-phenotype macrophages (F4/80+ CD86+ MHCII+) (**D**), activated NKT cells (NK1.1+CD69+) (**E**). Data are expressed as mean ± standard error (SEM). Data in (**A**–**E**) are analyzed with a one-way analysis of variance. * *p* < 0.05, ** *p* < 0.01, and *** *p* < 0.001; ns, not significant.

**Figure 3 cancers-16-00832-f003:**
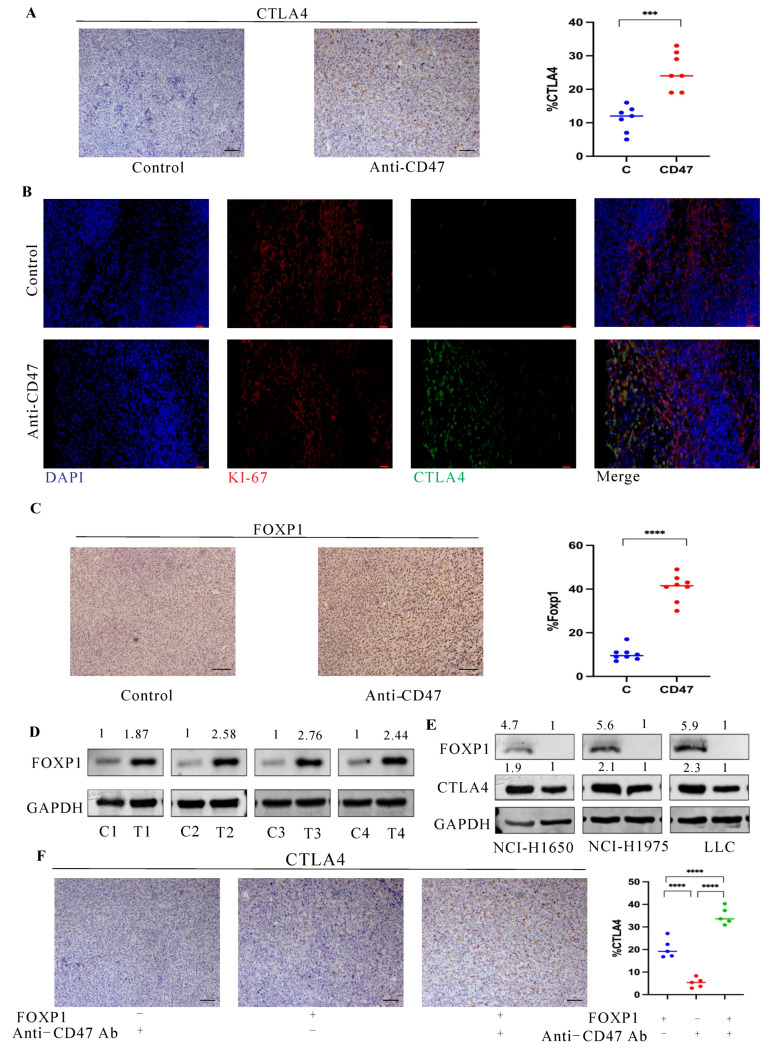
Effect of Foxp1 on CTLA4 up-regulation after CD47 blockade in non-small cell lung cancer (NSCLC) cells. (**A**) The fraction of CTLA4^+^ cells in tumor tissues of LLC cells was evaluated by immunohistochemistry (IHC) after treatment with anti-CD47 Ab or control IgG. The fraction of CTLA4^+^ cells was greater in the anti-CD47 Ab-treated group compared to the control IgG. (**B**) Immunofluorescence staining of ΚΙ-67 and CTLA4 in tumor tissues of LLC cells upon treatment with anti-CD47 Ab. Scale bar: 50 μm. (**C**) The fraction of Foxp1+ cells in tumor tissues of LLC cells, as assessed via IHC, after treatment with anti-CD47 Ab compared to control IgG. Scale bar: 50 μm. (**D**) Western blot analysis illustrates protein levels of FOXP1 after treatment with Anti-CD47 Ab (Control group: C1–4; treatment group with Anti-CD47 Ab: T1–4). (**E**) Western blot analysis for NCI-H1650, NCI-H1975, and LLC cells illustrates the protein levels of CTLA4 in response to FOXP1 depletion. The uncropped blots are shown in Appendix A. (**F**) After anti-CD47 Ab treatment, as evaluated via IHC, the levels of CTLA4 level were changed in the *Foxp1*^−/−^ subcutaneous transplantation tumor model compared to the wild-type. Scale bar: 50 μm. Data are presented as mean ± standard error (SE). Data in (**A**–**F**) are evaluated with unpaired Student’s *t*-test. *** *p* < 0.001, **** *p* < 0.0001.

**Figure 4 cancers-16-00832-f004:**
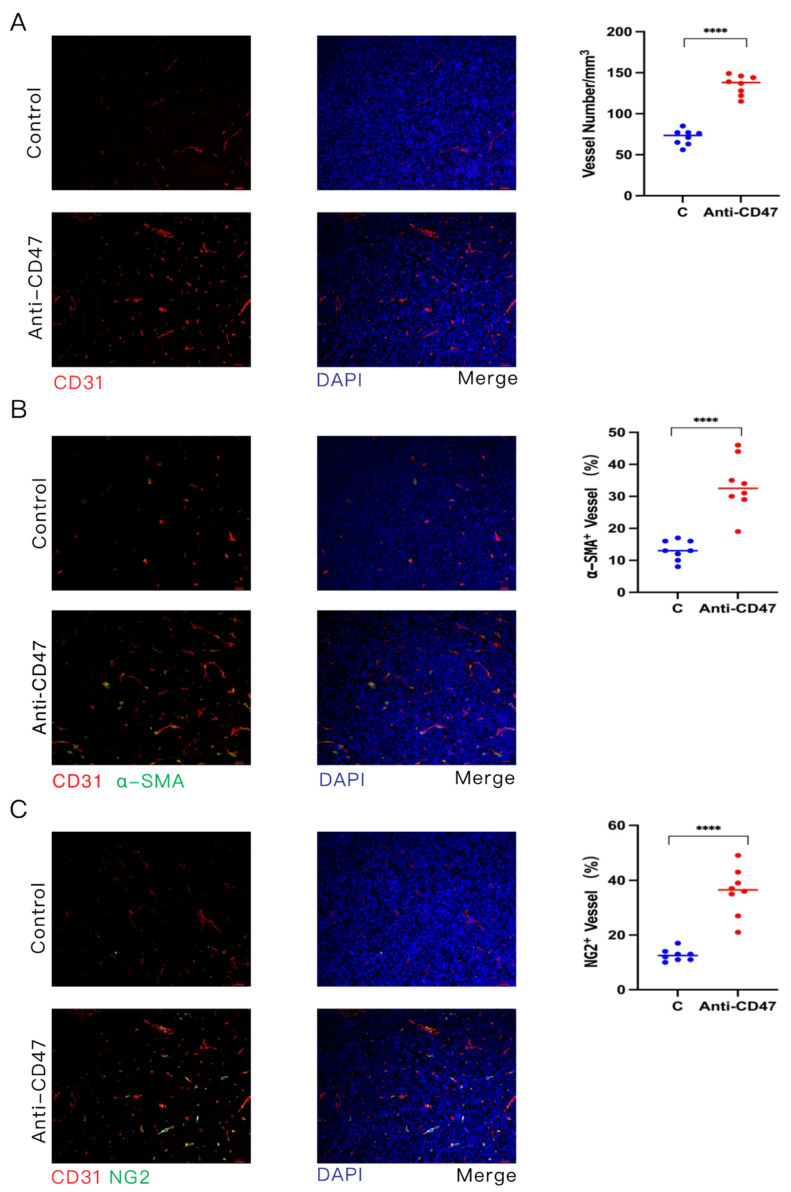
Effects of anti-CD47 antibody (Ab) on vascular maturation in tumor-bearing mice. Mice (*n* = 8) bearing tumors of subcutaneously transplanted LLC cells were treated with anti-CD47 Ab thrice weekly until tumors became palpable (60–80 mm^3^). Data are representative of at least three independent experiments. (**A**–**C**) Tumor tissues of LLC cells were collected on day 24 after treatment with anti-CD47 Ab. Blood vessel number within tumors as determined using CD31 staining (**A**). Images were obtained for blood vessels expressing α-SMA (**B**) and NG2 (**C**) in tumor tissues to evaluate blood vessels covered by wall cells and pericytes. Data are representative of three independent experiments. Scale bars, 20 μm. DAPI, 4′,6-diamidino-2-phenylindole. Data are expressed as mean ± standard error (SE). Data in (**A**–**C**) are evaluated with unpaired Student’s *t*-test. **** *p* < 0.0001.

**Figure 5 cancers-16-00832-f005:**
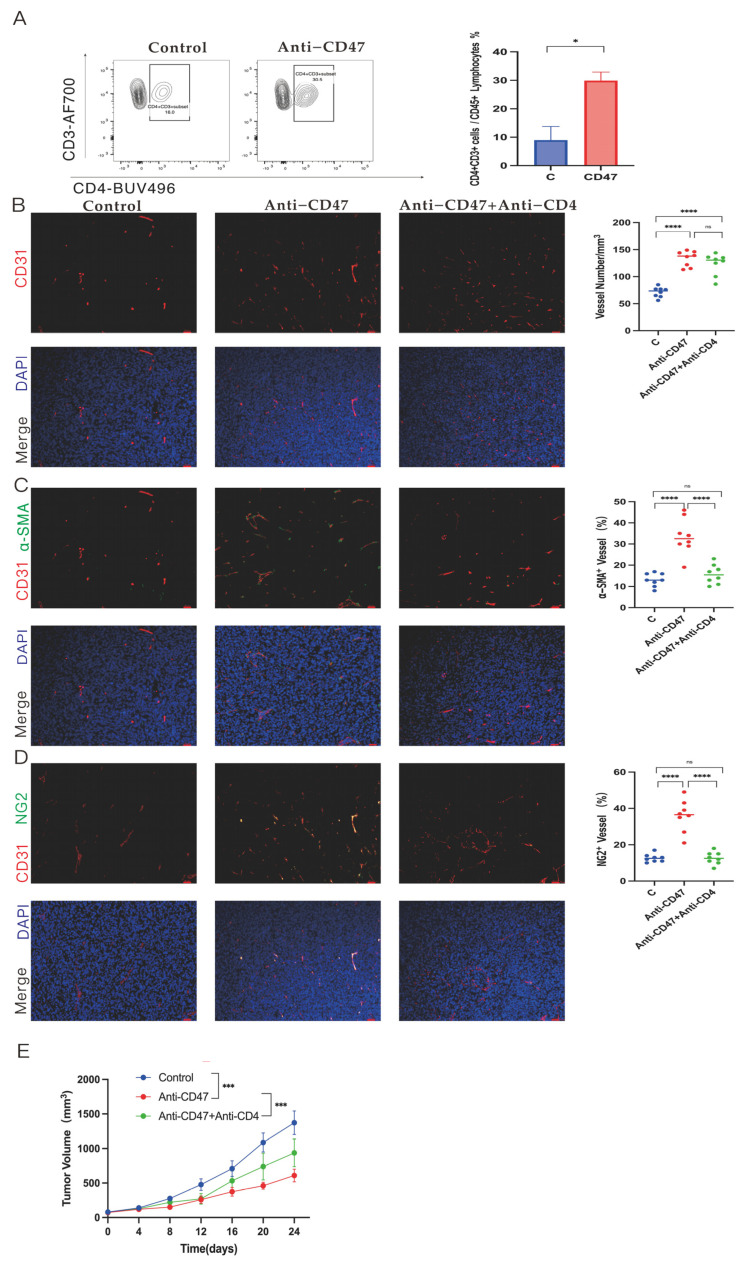
Role of CD4^+^ T cell infiltration by regulating tumor vascular function in the treatment of anti-CD47 antibody (Ab). (**A**) Flow cytometric analysis for CD4^+^ T cell infiltration in tumors of LLC cells implanted in C57BL/6 mice treated with control IgG or anti-CD47 Ab on day 24. The fraction of CD4^+^ T cells is changed by anti-CD47 Ab. (**B**–**D**) Representative immunofluorescence staining (CD31, α-SMA, NG2, and DAPI) of endothelial cells, wall cells, and pericytes in tumors of LLC cells treated with anti-CD47 Ab alone, combination therapy (anti-CD47 Ab and anti-CD4 Ab), and control IgG. Scale bar: 20 μm. (**E**) Subcutaneous tumor volume of LLC cells in C57BL/6 mice treated with control IgG, anti-CD47 Ab alone, or combination therapy. Data are representative of three independent experiments.Data are expressed as mean ± standard error (SE). Data in (**A**–**E**) are evaluated with a one-way analysis of variance. Data in (**A**) are evaluated with an unpaired Student’s *t*-test. * *p* < 0.05, *** *p* < 0.001, **** *p* < 0.0001; ns, not significant.

**Figure 6 cancers-16-00832-f006:**
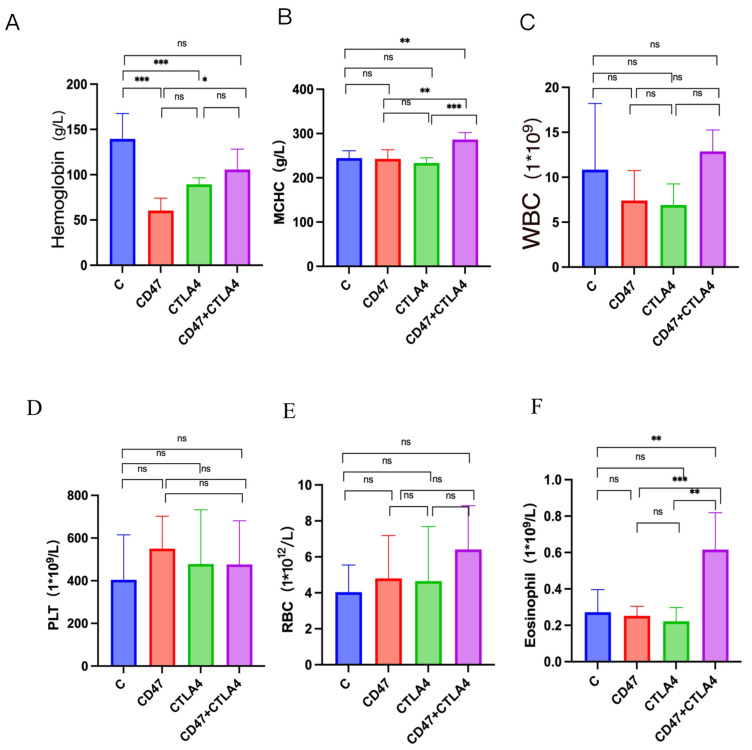
Role of CD47 and CTLA4 combined targeted therapy on blood cell changes in non-small cell lung cancer (NSCLC)-bearing mice. C57BL/6 mice with subcutaneous tumors of LLC cells were treated with anti-CD47 antibody (Ab) alone, anti-CTLA4 Ab alone, combination therapy, and control IgG three times per week for 24 days. Blood samples (0.5 mL) were retrieved from the eyeball extirpating in each group and analyzed using an automatic hemocytometer. Changes in hemoglobin (**A**), the mean corpuscular hemoglobin concentration (**B**) and white blood cells (**C**), red blood cells (**D**), platelets (**E**), eosinophils (**F**). Data are expressed as mean ± standard error (SE). Data in (**A**–**F**) are evaluated with a one-way analysis of variance. * *p* < 0.05, ** *p* < 0.01, and *** *p* < 0.001; ns, not significant.

## Data Availability

All study data are included in the article.

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
