# Peer review of "The Combination of Anti-CD47 Antibody with CTLA4 Blockade Enhances Anti-Tumor Immunity in Non-Small Cell Lung Cancer via Normalization of Tumor Vasculature and Reprogramming of the Immune Microenvironment"

_cancers, 2024, doi:10.3390/cancers16040832_

Round 1
Reviewer 1 Report
Comments and Suggestions for Authors
The manuscript proposed by Zhuang and collaborators entitled “Combination of anti-CD47 antibody with CTLA4 blockade enhances anti-tumor immunity in non-small cell lung cancer via normalization of tumor vasculature and reprogramming of the immune microenvironment” proposed a combination of anti-CD47 antibody with the anti-CTLA-4 agent in a murine model of NSCLC.
Even if the article presents interesting ideas, several problems are present in this manuscript, concerning results, discussion, and ultimately numerous images show incomplete statistical analysis.
For these reasons, I strongly suggest “major revision” of this manuscript, based on the following points.
Major points
Please eliminate the comments on the results in the figure legends.
1- Lane 22: what does the sentence “…..our study bridges the gap between immunotherapy and non-small cell lung cancer” mean?
2-fig. 2B-D: the authors reported that the infiltration of CD8+ exhausted T cells was reduced by the combination of anti-CD47 and anti-CTLA-4 antibodies, but in Figure 2B this data is lacking. After all, the statistical analysis between the combo and the anti-CTLA4 condition is lacking. The same analysis is lacking for Fig. 2C.
3- Lane 271: the sentence “No notable change was observed in tumor macrophage infiltration, either with combination therapy or monotherapy” is not correct as in Fig. S2A the amount of macrophage infiltration is higher in combo compared to control.
4- Lane 273: what it means with “…others”?
Lane 273-276. This sentence is incorrect, as the amount of MDSC, reported in Fig. S2 is not significantly changed, compared to anti-CTLA-4 treatment, so the authors cannot conclude that the combo reduced MDSC infiltrate.
In fig. S2C the authors reported a lack of increase of NK T cell population, whereas in Fig. 2E a significant increase of activated NK T cells was reported. The authors should explain this discrepancy.
In fig. 3F, lane 332-333, the authors reported that “…The data showed CTLA4 332 expression was reduced in Foxp1-deleted tumor model compared to WT tumor model”. This sentence is not correct as the results proposed in the figure are completely different. The authors employed LLC tumor cells Foxp1-/- and the corresponding WT. First of all, they should report the difference in Foxp1 In terms of IHC in both tumor models, and the CTLA4 modulation by anti.CD47 treatment in both tumor types. Finally, they should modify the corresponding graph.
Lane 388-390: this sentence is unfathomable, please correct or rewrite.
The results reported in Fig. S4 were already shown in S1C, please remove these data.
It is unclear why the authors explored the combination of anti-CD47 with anti-CD-4. Fig. 5 is incomplete: the effect of the anti-CD-4 antibody is lacking, so every conclusion is not correct.
I strongly suggest evaluating the angiogenesis after the combo of anti-CD-47 and anti-CTLA-4, as in terms of tumor shrinkage is a critical point and a possible further explanation of the effect of this combination.
In several parts of the Discussion, incomplete or incorrect results are reported and described.
I strongly suggest a full revision of the discussion, based on the new results obtained from the aforementioned points.
Minor points
1- Lane 50: please correct SIRRP with SIRP.
2- Lane 248-9: please correct the sentence “……However, in the monotherapy group and the combination therapy group, CD3+ T cell infiltration of the tumor infiltration of CD45+ immune cells”.
Lane 273: please correct “…was not observed” with “…were not observed”.
Lane 312: the expression “….CTLA4 expression was a significantly increase” should be corrected.
Lane 312-313: “…with immunoblotting,…”, please remove.
Lane 313: please correct “results” to “result”.
Lane 324 and 325, please correct "We” to “we” and remove whether, as it is repeated.
Fig. 3E: please define the samples in the blot panel.
Lane 428: please complete the sentence “To assess the impact of combined therapy targeting 428 CD47 and CTLA4 on blood cells of NSCLC-bearing mice”.
Lane 462: the expression “phagocytose tumors” is not fully correct. The macrophages can engulf tumor cells, not a tumor mass……..
Comments on the Quality of English LanguageAn extensive revision of the language is mandatory as the current version has many inaccuracies.
Author Response
|
Response to Reviewer 1 Comments
|
|||
|
|
1. Summary |
|
|
|
|
We have substantially updated the manuscript based on the valuable comments from the reviewers. Every effort has been made to significantly improve the manuscript. Following are the answers to reviewer’s questions point by point. The changes have been marked by red in the revised manuscript and indicated in the following. |
||
|
|
2. Questions for General Evaluation |
Reviewer’s Evaluation |
Response and Revisions |
|
|
Does the introduction provide sufficient background and include all relevant references? |
Yes |
|
|
|
Are all the cited references relevant to the research? |
Yes |
|
|
|
Is the research design appropriate? |
Must be improved |
|
|
|
Are the methods adequately described? |
Must be improved |
|
|
|
Are the results clearly presented? |
Must be improved |
|
|
|
Are the conclusions supported by the results? |
Must be improved |
|
|
|
3. Point-by-point response to Comments and Suggestions for Authors |
||
|
|
Comments 1: [Please eliminate the comments on the results in the figure legends.]
|
||
|
|
Response 1: Thank you for pointing this out. We agree with this comment. Due to the comments and inaccuracy on the results in the figure legends , we have made modifications to the following the figure legends respectively. 1- Lane 347: Figure 3 legend: Effect of Foxp1 on CTLA4 up-regulation after CD47 blockade in non-small cell lung cancer (NSCLC) cells..(page12) 2-Lane350: Figure 3B legend: Immunofluorescence staining of ΚΙ-67 and CTLA4 in tumor tissues of LLC cells upon treatment with anti-CD47 Ab.(page12) 3-Lane384: Effects of anti-CD47 antibody (Ab) on vascular maturation in tumor-bearing mice(page13) 4-Lane417: however. (page15) |
||
|
|
Comments 2: [1- Lane 22: what does the sentence “…..our study bridges the gap between immunotherapy and non-small cell lung cancer” mean?.] |
||
|
|
Response 2: Agree. The inaccurate expression has led to your misunderstanding of this sentence. We have, accordingly, modified“…..our study bridges the gap between immunotherapy and non-small cell lung cancer” as follows: Lane 23-25: our research provides compelling evidence supporting the efficacy of the anti-CD47 antibody and anti-CTLA4 antibody combination in treating NSCLC, thereby proposing meaningful avenues for future clinical trials addressing NSCLC.(page1)
Comments 3: [fig. 2B-D: the authors reported that the infiltration of CD8+ exhausted T cells was reduced by the combination of anti-CD47 and anti-CTLA-4 antibodies, but in Figure 2B this data is lacking. After all, the statistical analysis between the combo and the anti-CTLA4 condition is lacking. The same analysis is lacking for Fig. 2C. ] Response 3: Thank you for your review and positive comments on our work. We have modified the result description in Figure B:Lane269-271: the infiltration of exhausted CD8+ T cells was reduced in the combination therapy and anti-CTLA4 groups compared with the control group; however, such alterations were not observed in the anti-CD47 Ab groups(page7) .In addition,We have supplemented Figure2A-E statistical analysis.(Figure2 A-E,page8)
Comments 4: [Lane 273: what it means with “…others”? ] Response 4: As the reviewer asked, this sentence means combination therapy increased M1-phenotype macrophage (F4/80+ CD86+ MHCII+) infiltration in tumors, while such alterations was not observed in the monotherapy (anti-CD47 Ab and anti-C TLA4 Ab) and control group. “others “refers to the monotherapy (anti-CD47 Ab and anti-C TLA4 Ab) and control group.
Comments 5: [Lane 271:This sentence is incorrect, as the amount of MDSC, reported in Fig. S2 is not significantly changed, compared to anti-CTLA-4 treatment, so the authors cannot conclude that the combo reduced MDSC infiltrate. ] Response 5: As suggested by the reviewer, We have finished modifying the result description of FigS2. As follows:Lane279-283:Additionally, it was observed that the proportion of myeloid-derived suppressor cells (MDSCs) infiltration considerably decreased in the combination therapy and mono-therapy groups compared to the control group. Furthermore, in the treatment groups (combination and monotherapy), MDSCs were reduced in the combination group com-pared to the anti-CD47 group (Figure S2B). Comments 6: [In fig. S2C the authors reported a lack of increase of NK T cell population, whereas in Fig. 2E a significant increase of activated NK T cells was reported. The authors should explain this discrepancy. ] Response 6: Thank you for your valuable feedback. However, we would like to clarify that our article want to describe the number of activated NKT cells accounted for a higher percentage of the total number of NKT cells in combination therapy compared to the monotherapy groups and control group. It does not refer to the number of activated NKT cells. The following modifications are made:Lane288-291: Importantly, after combination therapy, the number of activated NKT cells accounted for a higher percentage of the total number of NKT cells,compared to the monotherapy groups and control group.
Comments 7:[In fig. 3F, lane 332-333, the authors reported that “…The data showed CTLA4 332 expression was reduced in Foxp1-deleted tumor model compared to WT tumor model”. This sentence is not correct as the results proposed in the figure are completely different. The authors employed LLC tumor cells Foxp1-/- and the corresponding WT. First of all, they should report the difference in Foxp1 In terms of IHC in both tumor models, and the CTLA4 modulation by anti.CD47 treatment in both tumor types. Finally, they should modify the corresponding graph.] Response 7: Thank you for your valuable comments. We are sorry that due to the errors of placing our subscripts and pictures, you may have misunderstood this sentence and the corresponding picture. we would like to clarify that the purpose of this experiment was to determine whether the expression of CTLA4 could be regulated through Foxp1 after anti-CD47 treatment. We have revised the pictures and subscripts.(as shown in fig.3F) In addition, we have added pictures to prove the difference in Foxp1 In terms of IHC in both tumor models.(as shown in fig.S3E)
Comments 8:[Lane 388-390: this sentence is unfathomable, please correct or rewrite.] Response 8: As suggested by the reviewer,this error has been corrected in the revised manuscript. [Lane 396-398: A recent report revealed that vascular normalization and immunostimulatory re-programming might regulate each other, highlighting the key role of infiltrating CD4 T cells in promoting tumor vascular normalization.]
Comments 9:[The results reported in Fig. S4 were already shown in S1C, please remove these data.] Response 9: As suggested by the reviewer, Fig. S4 has been removed.
Comments 10:[It is unclear why the authors explored the combination of anti-CD47 with anti-CD-4. Fig. 5 is incomplete: the effect of the anti-CD-4 antibody is lacking, so every conclusion is not correct.] Response 10: Thank you for your valuable feedback. A previous report revealed that CD4T cells play a crucial role in vessel normalization.(As showed in reference 26)we would like to clarify that the purpose of this experiment is whether anti-CD47 Ab regulated intra-tumor vascular normalization through increased CD4+ cell infiltration. Accordingly,we have applied combining anti-CD47 Ab with anti-CD4 Ab.
Comments 11:[I strongly suggest evaluating the angiogenesis after the combo of anti-CD-47 and anti-CTLA-4, as in terms of tumor shrinkage is a critical point and a possible further explanation of the effect of this combination.] Response 11: Thank you for your valuable comments. We have completed evaluation of the angiogenesis after the combo of anti-CD-47 and anti-CTLA-4.(As showed in fig.s4)
Comments 12:[In several parts of the Discussion, incomplete or incorrect results are reported and described.I strongly suggest a full revision of the discussion, based on the new results obtained from the aforementioned points.] Response 12: As suggested by the reviewer, we have modified the partial discussion based on the new results obtained from the aforementioned points for better understanding. Specifically details are in the manuscript discussion section.
Comments 13:[ Lane 50: please correct SIRRP with SIRP.]
Response 13: As suggested by the reviewer, this error has been corrected in the revised manuscript.(As showed in lane 51)
Comments 14:[ Lane 248-9: please correct the sentence “……However, in the monotherapy group and the combination therapy group, CD3+ T cell infiltration of the tumor infiltration of CD45+ immune cells”.] Response 14: As suggested by the reviewer, we have revised this sentence. [ Lane256-257: However, in both the monotherapy group and the combination therapy group, CD3+ T cell infiltration of the tumors increased,]
Comments 15:[Lane 273: please correct “…was not observed” with “…were not observed” .] Response 15: This error has been corrected in the modified manuscript. [Lane274:were not observed in the others.]
Comments 16:[Lane 312: the expression “….CTLA4 expression was a significantly increase” should be corrected.] Response 16: As suggested by the reviewer, we have revised this sentence. [Lane318-321:the level of CTLA4 was significantly increased after treatment with anti-CD47 Ab, compared to the wild-type group. This result further confirmed that anti-CD47 Ab in-creased CTLA4 levels in tumor cells.]
Comments 17:[Lane 312-313: “…with immunoblotting,…”, please remove.] Response 17: This error has been corrected in the modified manuscript. [Lane315]
Comments 18:[Lane 313: please correct “results” to “result”.] Response 18: As suggested by the reviewer, we have revised this error. [Lane316]
Comments 19:[Lane 324 and 325, please correct "We” to “we” and remove whether, as it is repeated.] Response 19: As suggested by the reviewer, we have modified this sentence. [Lane328]
Comments 20:[Fig. 3E: please define the samples in the blot panel.] Response 20: As suggested by the reviewer, we have modified. [As showed in Fig.3E] Comments 21:[Lane 428: please complete the sentence “To assess the impact of combined therapy targeting 428 CD47 and CTLA4 on blood cells of NSCLC-bearing mice”. ] Response 21: As suggested by the reviewer, we have revised this sentence. [Lane441-442: We evaluated the impact of combined therapy targeting CD47 and CTLA4 on the blood cells of NSCLC-bearing mice.]
Comments 22:[Lane 462: the expression “phagocytose tumors” is not fully correct. The macrophages can engulf tumor cells, not a tumor mass……..] Response 22: As suggested by the reviewer, we have revised this error. [Lane461]
|
||

Reviewer 2 Report
Comments and Suggestions for Authors
The article has a novel and interesting approach and could provide relevant information regarding how immunotherapy modifies the tumor microenvironment. However, it needs to be extensively reviewed and has ethical severe issues from the perspective of animal experimentation that need to be explained/corrected. Below, I detail the most relevant points:
1. First, the article needs significant editing, including grammar and run-on sentences.
2. The animal experimentation committee code approving to animal experimentation procedures used in this study is not included.
3. In the Materials and Methods section, it is mentioned that animals were euthanized by decapitation. This is not a humane procedure approved for animal euthanasia. A reasoned explanation justifying this method of euthanasia is needed.
4. For the erythrocyte lysis buffer, the commercial source and reference are not mentioned, or if it is homemade, the recipe should be provided.
5. For the antibodies used, specify the clone and the reference number from the commercial source.
6. Retro-orbital bleeding should only be performed under terminal anesthesia due to the severity of adverse effects that can occur with this technique, even in skilled hands. Justify the use of this technique in this case.
7. In each figure, specify the statistical method used.
8. In Figure 1, it is mentioned that n=5; however, in Figure 1B in green (anti-CTLA4), there are more than 5 mice.
9. Explain how there is a standard deviation in Figure 1A in blue (control) at day 24 when only one mouse is left out of the initial 5 mice.
10. The graph in Figure 1A should be extended to cover approximately 60 days, which is when the last mouse died.
11. The original images of the blots/gels should show the complete membrane, not just the cropped band, and specify the molecular weight of the molecular weight standard.
Comments on the Quality of English LanguageThe text has numerous typographical, writing, grammatical, and punctuation errors.
Author Response
|
Response to Reviewer 2 Comments
|
|||
|
|
1. Summary |
|
|
|
|
We have substantially updated the manuscript based on the valuable comments from the reviewers. Every effort has been made to significantly improve the manuscript. Following are the answers to reviewer’s questions point by point. The changes have been marked by red in the revised manuscript and indicated in the following. |
||
|
|
2. Questions for General Evaluation |
Reviewer’s Evaluation |
Response and Revisions |
|
|
Does the introduction provide sufficient background and include all relevant references? |
Can be improved |
|
|
|
Are all the cited references relevant to the research? |
Can be improved |
|
|
|
Is the research design appropriate? |
Must be improved |
|
|
|
Are the methods adequately described? |
Must be improved |
|
|
|
Are the results clearly presented? |
Can be improved |
|
|
|
Are the conclusions supported by the results? |
Must be improved |
|
|
|
3. Point-by-point response to Comments and Suggestions for Authors |
||
|
|
Comments 1: [1.First, the article needs significant editing, including grammar and run-on sentences.] |
||
|
|
Response 1: Thank you for pointing this out. We agree with this comment. We have revised and edited the article by blue.
|
||
|
|
Comments 2: [2.The animal experimentation committee code approving to animal experimentation procedures used in this study is not included.] |
||
|
|
Response 2: Thank you for pointing this out. We are apologized that you misunderstood this issue, because we did not provide The animal experimentation committee code in time. The experiment was approved by the Experimental Animal Ethics Committee of Fujian Medical University.(The animal experimentation committee code:IACUC FJMU 2 0 2 3-Y-0 5 3 8)
Comments 3: [3.In the Materials and Methods section, it is mentioned that animals were euthanized by decapitation. This is not a humane procedure approved for animal euthanasia. A reasoned explanation justifying this method of euthanasia is needed.] Response 3: Thank you for pointing out the shortcomings. We are apologized that You're confused about the ethics of animal testing. This reason is mainly the inaccurate expression and the inadequate description of the specific details of animal euthanasia. Below is our detailed description of the details of animal euthanasia: [Lane113-116:Mice were completely anesthetized with 2-3% isoflurane for 1-2 minutes before the experiment, causing the animals to rapidly lose consciousness. When the animal was dead, 2-3% isoflurane was continued for 1 minute, and then the animal was euthanized by cervical dislocation method. ]
Comments 4: [4.For the erythrocyte lysis buffer, the commercial source and reference are not mentioned, or if it is homemade, the recipe should be provided.] Response 4: As suggested by the reviewer, We have provided a commercial source of the erythrocyte lysis buffer. (Lane137:Purchased from BD Biosciences, Cat: 555899, Lot: 1250221)
Comments 5: [5.For the antibodies used, specify the clone and the reference number from the commercial source.] Response 5: As suggested by the reviewer, we have supplemented the clone and the reference number from the commercial source for the antibodies used. [ page3-4]
Comments 6: [6.Retro-orbital bleeding should only be performed under terminal anesthesia due to the severity of adverse effects that can occur with this technique, even in skilled hands. Justify the use of this technique in this case.] Response 6: Thank you for your valuable feedback. First all,we would like to clarify that our animal experiment operation is strictly in accordance with the experimental animal operation code(The animal experimentation committee code:IACUC FJMU 2023-Y-0538). Implementation steps of Retro-orbital blood: 1)Retro-orbital bleeding were be performed under terminal anesthesia. 2)Blood is collected using microhematocrit tubes to minimize risk of injury. 3)Only one eye may be sampled at any time.[If attempted collection from one eye is unsuccessful, an alternate method approved in the Animal Protocol (e.g. submandibular or saphenous route) must be used, rather than reattempting retro-orbital collection from the same or opposite eye.] 4)Alternate between left and right eyes per session. 5)No more than 1 collection performed per 7 days (alternate eyes). 14 days between collections in the same eye. 6)We consult veterinary staff prior to conducting experiments for demonstration and training of proper technique to reduce risk of trauma. 7)A maximum of 3 procedures may be performed per eye (up to 6 collections total). 8)If injury and/or rupture of the eye or surrounding tissues occurs due to this method, the animal must be immediately euthanized or an OAR veterinarian consulted for guidance on treating mice. 9)After collecting blood, the experimental mice were subjected to a topical ophthalmic anesthetic under the guidance of veterinarians. In addition, due to the inaccurate expression, we have modified some operational details of Retro-orbital bleeding experiment for better understanding. [page5,Lane195: Following that, 70-100ul of blood from the mouse was collected in microhematocrit tubes by Retro-orbital bleeding.]
Comments 7:[7. In each figure, specify the statistical method used.] Response 7: As suggested by the reviewer,we have added specify the statistical method used In each figure.[page6、8、10、12、14、16]
Comments 8:[8. In Figure 1, it is mentioned that n=5; however, in Figure 1B in green (anti-CTLA4), there are more than 5 mice.] Response 8: Thank you for your review and positive comments on our work. We are apologized that You're confused about this issue. We have corrected the error that you mentioned.
Comments 9:[9.Explain how there is a standard deviation in Figure 1A in blue (control) at day 24 when only one mouse is left out of the initial 5 mice.] Response 9: As suggested by the reviewer, there was a standard deviation,because the last two mice died on day 24 and day 27.
Comments 10:[10. The graph in Figure 1A should be extended to cover approximately 60 days, which is when the last mouse died.] Response 10: Thank you for your valuable feedback. As suggested by the reviewer,we have modified The graph in Figure 1A.
Comments 11:[11.The original images of the blots/gels should show the complete membrane, not just the cropped band, and specify the molecular weight of the molecular weight standard.] Response 11: As suggested by the reviewer, We have added the complete membrane for the blots and specified the molecular weight of the molecular weight standard.
|
||

Reviewer 3 Report
Comments and Suggestions for Authors
The manuscript entitled “Combination of anti-CD47 antibody with CTLA4 blockade enhances anti-tumor immunity in non-small cell lung cancer via normalization of tumor vasculature and reprogramming of the immune microenvironment” is well written and gives an interesting contribution to immunotherapy of non-small cell lung cancer, offering a good cue for future studies. Nevertheless, I would suggest some minor modifications to improve the presentation of the results and give more weight to the manuscript.
Line18-19: The treatments of non-small cell lung cancer have developed rapidly in recent years, and chemotherapy may be not a preferred therapeutic strategy in all patients.
Line120: “n =6-8” refers to the number of mice? However, “n = 5” showed in the Figure 1.
Line 222: Adding tumor picture in Figure 1 will be better.
Line 252/254: “Figure S1B” and “Figure S1C” is wrong, please check carefully.
Line 305: “upregulated in anti-CD47 Ab group compared with controls” may be better.
Line 339/412: Please describe the number of mice in all animal experiments.
Line 391: The chart of Figure S4A is wrong.
Line 441: Font size of Figure 6C should be revised.
Comments on the Quality of English LanguageMinor editing of English language required.
Author Response
|
Response to Reviewer 3 Comments
|
|||
|
|
1. Summary |
|
|
|
|
We have substantially updated the manuscript based on the valuable comments from the reviewers. Every effort has been made to significantly improve the manuscript. Following are the answers to reviewer’s questions point by point. The changes have been marked by red in the revised manuscript and indicated in the following. |
||
|
|
2. Questions for General Evaluation |
Reviewer’s Evaluation |
Response and Revisions |
|
|
Does the introduction provide sufficient background and include all relevant references? |
Can be improved |
|
|
|
Are all the cited references relevant to the research? |
Can be improved |
|
|
|
Is the research design appropriate? |
Yes |
|
|
|
Are the methods adequately described? |
Must be improved |
|
|
|
Are the results clearly presented? |
Yes |
|
|
|
Are the conclusions supported by the results? |
Yes |
|
|
|
3. Point-by-point response to Comments and Suggestions for Authors |
||
|
|
Comments 1: [Line18-19: The treatments of non-small cell lung cancer have developed rapidly in recent years, and chemotherapy may be not a preferred therapeutic strategy in all patients.] |
||
|
|
Response 1: Thank you for pointing this out. As suggested by the reviewer,we have revised this sentence.[page1,Lane18-19:Currently, the main therapeutic strategy for non-small cell lung cancer include: chemotherapy, targeted therapy, and immunotherapy. ]
|
||
|
|
Comments 2: [Line120: “n =6-8” refers to the number of mice? However, “n = 5” showed in the Figure 1.] |
||
|
|
Response 2: Thank you for your review and positive comments on our work. We are apologized that you misunderstood this issue, due to the number of mice is not consistent in context. We have corrected the above error.
Comments 3: [Line 222: Adding tumor picture in Figure 1 will be better.] Response 3: Thank you for your valuable feedback. However,due to the different time points of different experiments, it is difficult to dissect the tumor of all animal models at once. Therefore,tumor picture may not be available.
Comments 4: [Line 252/254: “Figure S1B” and “Figure S1C” is wrong, please check carefully.] Response 4: As suggested by the reviewer, we have revised. .
Comments 5: [Line 305: “upregulated in anti-CD47 Ab group compared with controls” may be better.] Response 5: As suggested by the reviewer, we have supplemented
Comments 6: [Line 339/412: Please describe the number of mice in all animal experiments.] Response 6: As suggested by the reviewer, we have supplemented.(n
Comments 7:[Line 391: The chart of Figure S4A is wrong.] Response 7: As suggested by the reviewer, we have modified.
Comments 8:[Line 441: Font size of Figure 6C should be revised.] Response 8: As suggested by the reviewer, we have modified.
|
||

Round 2
Reviewer 1 Report
Comments and Suggestions for Authors
The revised version of the manuscript entitled “The combination of anti-CD47 antibody with CTLA4 blockade 2 enhances anti-tumor immunity in non-small cell lung cancer 3 via normalization of tumor vasculature and reprogramming of 4 the immune microenvironment” is considerably improved compared to the first version, but some issues need to be solved.
Major points
Although some results have been eliminated from the figure legends, this is not complete. Please remove the results from all the figure legends (i.e. lanes 235-238…).
Fig. 2B: the statistical analysis reveals that the reduction of exhausted CD8+ T-Cells is mainly a consequence of the antiCTLA4 treatment, whereas in Fig. 2C the increase of NK 1.1 cells is related to the anti cD47 treatment. In both cases, the combo is not responsible for the reduction of exhausted CD8+ T cells and the increase of NK 1.1 cells. I suggest modifying the text, lanes 269-274.
In the reply to the Editor document, the authors reported that “In addition, we have added pictures to prove the difference in Foxp1 In terms of IHC in both tumor models (as shown in fig.S3E)”. This figure is not described in the manuscript.
In addition the authors reported in the Response 11: Thank you for your valuable comments. We have completed evaluation of the angiogenesis after the combo of anti-CD-47 and anti-CTLA-4(as showed in fig.s4). This is not true; this experiment has not been reported in Fig. S4 but in Fig. S5. The result should be described in the manuscript.
Author Response
|
Response to Reviewer 1 Comments
|
|||
|
|
1. Summary |
|
|
|
|
We have substantially updated the manuscript based on the valuable comments from the reviewers. Every effort has been made to significantly improve the manuscript. Following are the answers to reviewer’s questions point by point. The changes have been marked by red in the revised manuscript and indicated in the following. |
||
|
|
2. Questions for General Evaluation |
Reviewer’s Evaluation |
Response and Revisions |
|
|
Does the introduction provide sufficient background and include all relevant references? |
Yes |
|
|
|
Are all the cited references relevant to the research? |
Yes |
|
|
|
Is the research design appropriate? |
Can be improved |
|
|
|
Are the methods adequately described? |
Yes |
|
|
|
Are the results clearly presented? |
Can be improved |
|
|
|
Are the conclusions supported by the results? |
Yes |
|
|
|
3. Point-by-point response to Comments and Suggestions for Authors |
||
|
|
Comments 1: [Although some results have been eliminated from the figure legends, this is not complete. Please remove the results from all the figure legends (i.e. lanes 235-238…).]
|
||
|
|
Response 1: Thank you for pointing this out. We agree with this comment. We have made modifications to the following the figure legends respectively. 1- Lane 235-237: In non-small cell lung cancer (NSCLC) mouse models,tumor volume was presented after treatment with Anti-CD47 Ab,Anti-CTLA4 Ab or Anti-CD47 Ab plus Anti- CTLA4 Ab.(page7)
|
||
|
|
Comments 2: [Fig. 2B: the statistical analysis reveals that the reduction of exhausted CD8+ T-Cells is mainly a consequence of the antiCTLA4 treatment, whereas in Fig. 2C the increase of NK 1.1 cells is related to the anti cD47 treatment. In both cases, the combo is not responsible for the reduction of exhausted CD8+ T cells and the increase of NK 1.1 cells. I suggest modifying the text, lanes 269-274..] |
||
|
|
Response 2: Thank you for your review and positive comments on our work. We only describe the results of the experiment and do not conclude that combination therapy is responsible for the reduction of depletion of CD8 T cells and the increase of NK 1.1 cells. In addition, there was a tendency for anti-CD47 antibody treatment to improve NK 1.1 cells, but this trend was not statistically significant. Therefore, we believe that it may not be reasonable to describe the enhancement of NK 1.1 cell infiltration by anti-CD47 antibody therapy.
Comments 3: [In the reply to the Editor document, the authors reported that “In addition, we have added pictures to prove the difference in Foxp1 In terms of IHC in both tumor models (as shown in fig.S3E)”. This figure is not described in the manuscript..] Response 3: Thank you for your valuable feedback. We have added the description of fig.S3E to the manuscript.(Lane:333-337)
Comments 4: [In addition the authors reported in the Response 11: Thank you for your valuable comments. We have completed evaluation of the angiogenesis after the combo of anti-CD-47 and anti-CTLA-4(as showed in fig.s4). This is not true; this experiment has not been reported in Fig. S4 but in Fig. S5. The result should be described in the manuscript..] Response 4: Thank you for your valuable comments. We have modified the serial number of the pictures and described the results of the pictures in the manuscript.(Lane382-387)
4. Response to Comments on the Quality of English Language |
||
|
|
|
||
|
|
Response 1: English language fine. No issues detected. |
||
|
|
5. Additional clarifications |
||
|
|
[Here, mention any other clarifications you would like to provide to the journal editor/reviewer.] |
||

Reviewer 2 Report
Comments and Suggestions for Authors
In answers 10 and 11 they say that they have modified the figure and that you show the complete membrane of the blots, but however that modification has not been made. Once this issue is corrected, the paper could be published.
Author Response
|
Response to Reviewer 2 Comments
|
|||
|
|
1. Summary |
|
|
|
|
We have substantially updated the manuscript based on the valuable comments from the reviewers. Every effort has been made to significantly improve the manuscript. Following are the answers to reviewer’s questions point by point. The changes have been marked by red in the revised manuscript and indicated in the following. |
||
|
|
2. Questions for General Evaluation |
Reviewer’s Evaluation |
Response and Revisions |
|
|
Does the introduction provide sufficient background and include all relevant references? |
Yes |
|
|
|
Are all the cited references relevant to the research? |
Yes |
|
|
|
Is the research design appropriate? |
Can be improved |
|
|
|
Are the methods adequately described? |
Can be improved |
|
|
|
Are the results clearly presented? |
Can be improved |
|
|
|
Are the conclusions supported by the results? |
Can be improved |
|
|
|
3. Point-by-point response to Comments and Suggestions for Authors |
||
|
|
Comments 1: [In answers 10 and 11 they say that they have modified the figure and that you show the complete membrane of the blots, but however that modification has not been made. Once this issue is corrected, the paper could be published.] |
||
|
|
Response 1: Thank you for pointing this out. We agree with this comment. We have showed the complete membrane of the blots. As followed:
Figure3.D
Foxp1: 75 kDa
GAPDH:37kDa
Figure3.E
Foxp1:75kDa
CTLA4:37kDa
GAPDH:37kDa
Supplement Figure3.C CTLA4:37kDa GAPDH:37kDa
Supplement Figure3.E CTLA4:37kDa |
||
|
|
GAPDH:37kDa |
||
|
|
4. Response to Comments on the Quality of English Language |
||
|
|
|
||
|
|
Response 2: I am not qualified to assess the quality of English in this paper. |
||
|
|
5. Additional clarifications |
||
|
|
[Here, mention any other clarifications you would like to provide to the journal editor/reviewer.] |
||

Round 3
Reviewer 1 Report
Comments and Suggestions for Authors
I strongly invite the authors to eliminate the result comments from Figs. 2-3-5-6.
Author Response
For research article
|
Response to Reviewer 1 Comments
|
|||
|
|
1. Summary |
|
|
|
|
We have substantially updated the manuscript based on the valuable comments from the reviewers. Every effort has been made to significantly improve the manuscript. Following are the answers to reviewer’s questions point by point. The changes have been marked by red in the revised manuscript and indicated in the following. |
||
|
|
2. Questions for General Evaluation |
Reviewer’s Evaluation |
Response and Revisions |
|
|
Does the introduction provide sufficient background and include all relevant references? |
Yes |
|
|
|
Are all the cited references relevant to the research? |
Yes |
|
|
|
Is the research design appropriate? |
Yes |
|
|
|
Are the methods adequately described? |
Yes |
|
|
|
Are the results clearly presented? |
Must be improved |
|
|
|
Are the conclusions supported by the results? |
Yes |
|
|
|
3. Point-by-point response to Comments and Suggestions for Authors |
||
|
|
Comments 1: [I strongly invite the authors to eliminate the result comments from Figs. 2-3-5-6..] |
||
|
|
Response 1: Thank you for pointing this out. We agree with this comment. We have eliminated the result comments from Figs. 2-3-5-6 respectively. 1- Lane 293-297(page10) 2- Lane 345-351(page12) 3- Lane 422-426(page16) 4- Lane 452-454(page17) |
||
|
|
|
||
|
|
4. Response to Comments on the Quality of English Language |
||
|
|
|
||
|
|
Response 1: English language fine. No issues detected. |
||
|
|
5. Additional clarifications |
||
|
|
[Here, mention any other clarifications you would like to provide to the journal editor/reviewer.] |
||

Round 4
Reviewer 1 Report
Comments and Suggestions for Authors
The final version is now ready for publication in the Cancers Journal
Author Response
|
Response to Reviewer 1 Comments
|
|||
|
|
1. Summary |
|
|
|
|
We have substantially updated the manuscript based on the valuable comments from the reviewers. Every effort has been made to significantly improve the manuscript. Following are the answers to reviewer’s questions point by point. The changes have been marked by red in the revised manuscript and indicated in the following. |
||
|
|
2. Questions for General Evaluation |
Reviewer’s Evaluation |
Response and Revisions |
|
|
Does the introduction provide sufficient background and include all relevant references? |
Yes |
|
|
|
Are all the cited references relevant to the research? |
Yes |
|
|
|
Is the research design appropriate? |
Yes |
|
|
|
Are the methods adequately described? |
Yes |
|
|
|
Are the results clearly presented? |
Yes |
|
|
|
Are the conclusions supported by the results? |
Yes |
|
|
|
3. Point-by-point response to Comments and Suggestions for Authors |
||
|
|
Comments 1: [The final version is now ready for publication in the Cancers Journal.] |
||
|
|
Response 1: Thank you for putting forward many valuable suggestions for this paper.
|
||
|
|
|
||
|
|
4. Response to Comments on the Quality of English Language |
||
|
|
|
||
|
|
Response 1: English language fine. No issues detected. |
||
|
|
5. Additional clarifications |
||
|
|
[Here, mention any other clarifications you would like to provide to the journal editor/reviewer.] |
||
